# Is Every Vein a Real Vein? Cross-Section of the Wing of *Matsucoccus Pini* (Insecta, Hemiptera, Coccoidea: Matsucoccidae)

**DOI:** 10.3390/insects14040390

**Published:** 2023-04-17

**Authors:** Barbara Franielczyk-Pietyra, Małgorzata Kalandyk-Kołodziejczyk, Jowita Drohojowska

**Affiliations:** Institute of Biology, Biotechnology and Environmental Protection, University of Silesia in Katowice, Bankowa 9, 40-007 Katowice, Poland; barbara.franielczyk-pietyra@us.edu.pl (B.F.-P.); malgorzata.kalandyk@us.edu.pl (M.K.-K.)

**Keywords:** archaeococcoids, cross-section, wings, Matsucoccidae, wing veins

## Abstract

**Simple Summary:**

Wings play an important role in insect locomotion. It is the venation that provides reinforcement to the wings. An important question is whether the lines considered to be veins are visible in cross-section or are they just the folds of the wing’s membrane. The cross-sections of the forewing of the scale insect *Matsucoccus pini* were cut and confirmed the presence of only one vein—radius. There were also campaniform sensilla described, for the first time, on both the dorsal and ventral sides of the wing base.

**Abstract:**

Wings of *Matsucoccus pini* males were studied. Using light and scanning electron microscopes, both sides of the wing membrane, dorsal and ventral, were examined. The presence of only one vein in the common stem was confirmed by the cross-section, namely the radius. The elements regarded as subcostal and medial veins were not confirmed as veins. On the dorsal side of the wings, a cluster of campaniform sensilla is shown for the first time in the family Matsucoccidae, through SEM, and two additional sensilla were found on the ventral side. There was a lack of alar setae, microtrichia as well as pterostigma. This is the second cross-section of the wing among scale insects. We propose the following nomenclature for the wings in the family Matsucoccidae: subcostal thickening (sct), radius (R), median fold (med) and anal fold (af).

## 1. Introduction

Scale insects (Hemiptera, Coccomorpha) constitute the superfamily Coccoidea Fallén, 1814, composed of over 8500 species [1]. These insects prefer a warm and dry climate. The number of species decrease to the north and northeast. As other Sternorrhyncha Amyot et Audinet-Serville, 1843, they are sap-sucking plant parasites. They are very closely associated with the host plants and, at the same time, are regarded as the worst plant pests. Their harmfulness to plants results from the direct sucking of sap, but also from the transmission of microbial diseases. Coccoids are more diverse than any other Sternorrhynchan taxa, considering evolutionary lineages (families), species richness, morphology and genetic systems [2,3]. Females are closely associated with the host plant, while winged males can fly between plants.

Usually, scale insects are divided into two informal groups: archaeococcoids and neococcoids. The former comprises around 700 species, while the latter one a significantly higher number of species, with over 7000 species, as noted by various authors [2,3,4]. Some authors recognize two superfamilies—Orthezioidea Amyot et Serville, 1843 and Coccoidea. Various authors place Coccoidea within the infraorder or suborder Coccinea Fallén, 1814 [5,6]. It can be challenging to determine plesiomorphic characters for all species classified in these two groups. According to Gullan and Cook [2], archaeococcoids possess certain plesiomorphic features, such as abdominal spiracles and mostly compound eyes in males. However, according to Gavrilov-Zimin [6], not all archaeoccoids have abdominal spiracles, for example, *Phenacoleachia* Cockerell, 1899, some species of *Kuwania* Cockerell in Fernald, 1903, *Porphyrophora* Brandt in Brandt and Ratzeburg, 1833 and even some Ortheziidae lost abdominal spiracles. According to some authors [2,3,4], neococcoids constitute a monophyletic group that is characterized by features, including paternal genome elimination (PGE) and the absence of abdominal spiracles. In their studies, Gavrilov-Zimin and Danzig [7] and Danzig and Gavrilov-Zimin [8] indicate, that not all neococcoids have PGE, as evidenced by species *Stictococcus* Cockerell, 1903 or *Lachnodius* Maskell, 1896. Moreover, only about 6% of scale insects have been studied cytogenetically until now [8].

There is extreme sexual dimorphism among scale insects—adult females do not possess wings and their bodies are sac-like with fused body parts, usually strongly covered in wax [9].

According to Koteja [10], the wing absence in the female is the result of the habitat of ancestral scale insects. There is another hypothesis that the absence of wings in scale insect females is an adaptation to their parasitic life behavior, i.e., feeding on plants [6,11].

In contrast, adult males have their bodies divided into head, thorax and abdomen, are usually winged, and do not possess functional mouthparts [9]. Males have only one pair of functional membranous wings, while the second pair is reduced to hamulohalteres [12]. The venation of wings among these insects is highly reduced. The interpretation of the wing‘s venation terminology in scale insects varies among researchers. In recent years, the nomenclature of the wings of male scale insects has been under discussion and a new nomenclatural system for wing venation in coccoids has been proposed [13].

One of the archaeococcoid families is Matsucoccidae Morrison, 1927 comprised only two genera—*Eomatsucoccus* Koteja, 1988 and *Matsucoccus* Cockerell, 1909. The genus *Matsucoccus* was previously included in the family Margarodidae Morrison, 1928 [14], but currently is mostly considered as a representative of a separate family—the Matsucoccidae [15,16,17]. The genus *Matsucoccus* consists of six fossils and thirty-three extant species [1].

The representatives of the Matsucoccidae family are distributed throughout the Northern Hemisphere, feed only on pines and some are considered serious pests [18]. The only species reported to date from Poland is *Matsucoccus pini* Green, 1925 (Hemiptera, Coccomorpha) [1].

Adult males of the Matsucoccidae are known to possess elongated, mostly narrow wings with a peculiar pinnate sculpture, which is considered unique among other members of Sternorrhyncha [19].

Morrison [14], Beardsley [20] and recently Gavriliov-Zimin [6] considered the current Matsucoccidae as an old and primitive group, sharing numerous features with aphids. Koteja [15], on the other hand, suggested that *Matsucoccus* is one of the most specialized genera among archaeococcoids, characterized by derived features, such as the structure of the mouthparts and morphology of larval instars.

Currently, only adult males of certain extant species of *Matsucoccus* have been described, namely: *Matsucoccus bisetosus* Morrison, 1939, *M. feytaudi* Ducasse, 1941, *M. josephi* Bodenheimer et Harpaz, 1955 and *M. pini* [16,18,20,21,22]. Additionally, males of five fossil species have been described from the inclusions in Baltic amber [19]. All described males’ wings are well-developed except for the fossil species *Matsucoccus apterus* Koteja, 1984 [19], which is wingless.

Authors used various terminologies to describe the veins in Matsucoccidae [14,16,18,20,21,22]. Due to the confusing morphology and nomenclature, we decided to check the internal structure of the wing of *M. pini* to confirm the presence of veins in it. Worth mentioning is the nature of veins—these strengthening elements of the wings can comprise tracheae and hemolymph, but these elements are not always present inside the vein [23,24].

The only way to determine the course and the nature of the wing’s veins is via cross-section. As an attempt to check the internal structure of veins, the cross-section of the wings of Coccoidea was made for only one species so far—*Orthezia urticae* (Linnaeus, 1758) [25].

## 2. Materials and Methods

Samples

Male cocoons were collected from the bark crevices of *Pinus sylvestris* in May 2014 and 2016 in Katowice, Przystań and Kuźnia Raciborska (Southern Poland). Cocoons were held in the laboratory at room temperature until adult males emerged. After emerging, the insects were immersed in 70% alcohol for preservation.

Scanning electron microscopy

For SEM, specimens kept in 70% ethanol were dehydrated in a series of ethanol 80%, 90% and 96% for ten minutes in each concentration and 3 × 10 min in 100% ethanol [26]. Samples were mounted on holders, sputter-coated with gold and examined using a scanning electron microscope (Phenom XL, Phenom-World BV, The Netherlands) in the Scanning Electron Microscopy Laboratory at the Institute of Biology, Biotechnology and Environmental Protection, University of Silesia.

Light microscopy

Insect wings were placed on a Petri dish with 70% alcohol. Photographs were taken under a Nikon SMZ1500 stereomicroscope.

Histology

Specimens collected in 70% ethanol were transferred to 2.5% glutaraldehyde in a 0.05 M cacodylate buffer (pH 7.4). After washing in 0.1 M phosphate buffer (pH 7.4), the material was postfixed for 2 h using 1% OsO_4_ in phosphate buffer, dehydrated in a graded series of ethanol replaced by acetone and then embedded in an Epoxy Embedding Medium Kit (Sigma, St. Louis, MO, USA). Semithin sections were cut from the root to the tip of the wing with a diamond knife, on a Leica Ultracut UCT ultramicrotome, at the Institute of Biology, Biotechnology and Environmental Protection, University of Silesia. Each section had a thickness of 700 nm and was stained with methylene blue. Sectional cuts were analyzed using a Nikon Ni-U light microscope and photographed with a Nikon DS-Fi2 camera. The whole wing was cut into about 800 semithin sections but only 9 slices were selected. On the photos, they are aligned in order—costal margin at the top, anal margin at the bottom, upper surface to the left.

## 3. Results

The wings are well developed, narrow, membranous and large, almost as long as the body, with a broadly rounded apex (Figure 1).

Under a light microscope, the wing membrane is semitransparent except for the subcostal thickening and radius vein, which is intensely pigmented.

The wing surface studied under an electron scanning microscope is covered with a specific pinnate sculpture formed by parallel ripples with fine cuticular strips (Figure 2a). These numerous parallel ripples radiate from both sides of the median fold, anteriorly, reaching the radial vein and posteriorly wing margin and anal fold. Short parallel ripples arise also from both sides of the subcostal thickening and reach the anterior wing margin and radial vein. The ripples are irregular in the area of the anal lobe. Ripples are visible on both sides of the wing (Figure 2a,b).

The longitudinal subcostal thickening (sct) runs parallel to the anterior wing margin and extends almost to the wing’s apex. At the end of its course, it turns backward and ends more or less in the middle of the distal wing margin (Figure 3, Figure 4, Figure 5 and Figure 6).

Cross-sections of the analyzed wings are thicker at the beginning and become thinner at the end. The well developed and strongly pigmented radial vein (R) runs parallel to the subcostal thickening and terminates at about the middle length of the wing (Figure 3, Figure 4 and Figure 5a,b).

The costal area is distinctly sclerotized in its base. The area anterior to the subcostal thickening is narrow. The distance between the subcostal thickening and wing margin, as well as between the sct and radial vein, is almost the same. The bases of the subcostal thickening and radial vein are not fused. There is no pterostigma. The media vein is represented only by indentation (visible from the dorsal wing side). The base of the media vein does not seem to be connected with the radial vein. It can be seen as a separate element, parallel to R for the short distance from the base and then, running diagonally towards the distal wing margin. The media’s indentation is well visible up to about 4/5 of its length and its distal part can be seen as a bright band (Figure 4b, Figure 5 and Figure 6a).

The narrow fold of the anal lobe on the proximal hind margin of the wing provides the structure for holding the hamuli of the second pair of wings (hamulohalteres). The anal fold extends diagonally from the wing base to the posterior margin of the wing and delimits the anal lobe (Figure 1 black arrow and Figure 2b). Alar setae, as well as microtrichia, are absent. The cluster of eight campaniform sensilla is present on the dorsal surface of the wing, near the base of the subcostal thickening (Figure 7a,c). Two campaniform sensilla are present on the ventral side of the basal part of the subcostal thickening (Figure 7b,d).

## 4. Discussion

There is no other scale insect structure with such disputable descriptions as the wing’s venation, according to Koteja [27]. The nomenclature of the veins within the family Matsucoccidae varies a lot in interpretation [13,14,16,18,19,20,21,27,28]. To highlight the inconsistency in nomenclature, Table 1 presents the different terminologies used for the veins and folds of the forewings of *Matsucoccus*. Additionally, to broaden the discussion, the terminology of wing veins used by Morrison [14] for the wings of Margarodidae s.l., including *Matsucoccus* was also presented.

The presence of the costal vein was not observed on the cross-section during the present study. Our results are consistent with those produced for *Orthezia urticae* by Franielczyk-Pietyra et al. [25]. The costal vein has not been recognized in wings of scale insects so far, e.g., [16,27,29,30].

After the cross-section, the presence of a vein within the subcostal thickening was not observed, as suggested by Beardsley [20]. Morrison [14] instead indicated that the costal complex of the wing of *Matsucoccus* is composed of two well-developed veins, namely subcostal and radius. The presence of a subcostal in the forewing of *Matsucoccus pini* was suggested by Siewniak [21], as well as in the description of the wings of *Matsucoccus feytaudi* by Foldi [18]. Our results are also not consistent with the terminology proposed by Wu and Xu [13], who recognized the subcostal thickening as Subcosta in the wing of Matsucoccus bisetosus. 

The apical diagonal vein (RS) recognized by Morrison [14] in Margarodidae s.l. was not observed in *Matsucoccus pini*. According to Shcherbakov [31], this vein is lost in Matsucoccidae.

We observed only one vein on our cross-sections, which is the radial vein, most probably. This vein is well distinguished by the pronounced lumen. It runs below the subcostal thickening from the base of the wing towards its apex (Figure 1, Figure 3, Figure 4 and Figure 5a,b). Our results are different from those obtained by Franielczyk-Pietyra et al. [25], where two veins were observed in the subcostal ridge—the subcostal and radial veins, running together in a common stem. The subcostal ridge *sensu* Koteja [32] was interpreted as a well-developed radius by Hodgson and Foldi [16]. Generally, the term “subcostal ridge” has not been applied in the descriptions of males of *Matsucoccus*, e.g., [20,21], but Koteja used it to describe fossil species *Matsucoccus saxonicus* Koteja, 1986 [30]. The radial vein in the wings of *Matsucoccus* was described by other authors [13,16,18,20,21]. In the SEM picture, the radial vein can be seen as only slightly shorter than the subcostal thickening, reaching almost the apex of the wing. However, the cross-section clearly shows that the radial vein is considerably shorter and terminates approximately in the middle of the wing’s length.

It seems that the presence of two veins in the common stem is not a feature shared by all representatives of Coccoidea. However, the presence of two veins (Sc and R) in a common stem was considered a common feature shared by scale insects with their likely ancestors, Naibiidae [31]. On the other hand, the presence of one radial vein is characteristic of representatives of the Matsucoccidae. In other scale insect families, two veins (or stems of two veins) are recognized [13].

The media vein is merely an indentation, visible as a concave fault in the cross-section (Figure 4b, Figure 5 and Figure 6a). Beardsley [20] suggested that the media vein could be a fold or weak vein. It is actually not a vein, contrary to the descriptions of Siewniak [18,21], Hodgson and Foldi [16] and Wu and Xu [13]. Koteja [27,33], on the other hand, suggested that there is a complete reduction of the posterior ridge (cubital ridge) (“media vein” *sensu* Beardsley) [20] in the wing of *Matsucoccus*. A similar condition was observed in the wing of *Xylococcus* Lôw, 1882 (Xylococcidae Pergrande, 1898), which is a close relative of *Matsucoccus*, with a vestigial posterior ridge, that can be seen as a slightly sclerotized patch. Koteja [27] proposed the term “first light line” (MS), after Morrison [14], to describe the media’s indentation in the forewing of *Matsucoccus.* On the other hand, Shcherbakov [31] suggested that this vein is lost in the wings of Matsucoccidae (described in his paper as CuA). In the SEM image, the media’s indentation resembles a structure similar to a vein in its middle part. The cross-section shows that there is indentation throughout the whole course.

Our results are different from those obtained by Choi et al. [34] who recognized a costal complex and two main veins extending from the wing base to its apex in *Matsucoccus matsumurae* (Kuwana, 1905). The anal fold can be seen as a faint trace (Figure 1). It is not a vein, which is consistent with descriptions of different authors [13,16,18,20].

Taking into account the results of the present study, we propose the following nomenclature for the wings in the family Matsucoccidae: subcostal thickening (sct), radius (R), median fold (med) and anal fold (af).

Koteja [35] recognized apomorphic wing features in representatives of the Matsucoccidae, comprising the shape of the wing (narrow and parallel-sided), the presence of a very small anal field and a cluster of campaniform sensilla at the wing base as well as the absence of alar setae. The venation of archaeococcoids is more complex than that of neococcoids, according to many authors, e.g., [16].

Koteja’s research revealed the existence of a small group of campaniform sensilla (5–10 structures) dorsally, in the region of the costal complex near the wing base of fossil Matsucoccidae [19,28,34]. According to Foldi [18] and Hodgson and Foldi [16], there is a group of 7–9 circular sensoria near the base of the subcostal thickening in the wings of *Matsucoccus feytaudi* and *M. josephi*. Interestingly, the presence of these sensilla in the wings of *Matsucoccus* males was not mentioned by Morrison [14], Beardsley [20], Rieux [22] or Siewniak [21]. Due to the small number of sensilla and lack of wax, it was possible to count these small structures, and through our observation, we were able to confirm that there are eight campaniform sensilla forming only one cluster dorsally, on the base of subcostal thickening (Figure 7a,c). Our research confirms that the presence of these sensilla near the wing base is a distinctive feature of the genus *Matsucoccus.* Additionally, we also observed two campaniform sensilla on the ventral surface of the basal part of sct (Figure 7b,d).

According to Koteja [27], the presence of only one cluster of campaniform sensilla at the wing base is known exclusively in the Matsucoccidae family. However, both cluster and row of these sensilla occur in Xylococcidae, which are considered to be closely related to Matsucoccidae [27]. Furthermore, a row of these elements occurs in other archaeococcoids and primitive neococcoids [27]. In archaeococcoids, the number of wing sensilla may be much greater than in *Matsucoccus*, e.g., in *Stigmacoccus* sp. (Stigmacoccidae), where over 70 sensilla were noticed [16]. Alar sensilla in the family Matsucoccidae are shown for the first time using SEM in the present study.

In the present study, SEM photographs of the sensilla in the family Matsucoccidae were obtained for the first time, and to the best of our knowledge, such photographs have been only provided for *Orthezia urticae* by Franielczyk-Pietyra et al. [25].

We did not observe alar setae, which is consistent with descriptions of other species of Matsucoccidae [16,18,19,28]. Alar setae are present among sensilla in different taxa of archaeococcoids, as well as on the wing base of neococcoids [27]. The presence or absence of alar setae and sensilla were considered taxonomically significant by Hodgson and Foldi [16]. The absence of alar setae in Matsucoccidae raises a question about their significance in other families and their potential role in coccids flight and flight stability. Further research could investigate the distribution and density of alar setae across different taxa.

The presence of microtrichia was not observed either during the present study. Microtrichia are absent in most archaeococcoids, but they are present in the families, such as Phenacoleachiidae Cockerell, 1900, Pityococcidae McKenzie, 1942, Putoidae Beardsley, 1969 and Steingeliidae Morrison, 1927 [16], which are considered to belong either to archaeococcoids [2] or neococcoids [36]. Microtrichia are common among neococcoids [27].

The Matsucoccidae do not possess a pterostigma, e.g., [16,18,19,28]. The club-shaped pterostigma is present in some extant archaeococcoid families, e.g., Margarodidae, Monophlebidae Signoret, 1875 and Quinococcidae Wu, 2022 [13,27] and also in fossil families, e.g., Weitschatidae Koteja, 2008 and Jersicoccidae Koteja 2000 [33,37], but is absent in neococcoids [27].

The appearance and number of veins in coccoid males may differ among species within the same genus, according to Lambdin [38]. On the contrary, based on the descriptions of wings of both extant and fossil species, as well as our studies, we can conclude that the vein pattern is stable in the representatives of the genus *Matsucoccus*.

*Orthezia urticae* and *Matsucoccus pini* are assigned to archaeococcoids, but their wing venation pattern is different. There are two veins (Sc + R) in *Orthezia urticae*, whereas *Matsucoccus pini* possesses only one vein (R). Further studies are needed to check if the presence or absence of Sc is characteristic feature in different taxa of Coccoidea. Understanding the taxonomic significance of venation patterns and their correctness is necessary to establish the most accurate classification of scale insects, as well as other insects.

## 5. Conclusions

The research in question concerned the venation of the wings of a representative of scale insects, *Matsucoccus pini*. Studies were performed using both scanning electron microscopy and histological embedding techniques. The epon-embedded wing was cut into sections using a microtome. The analysis of 800 ultra-thin sections allowed us to choose nine slices, on which the course of venation of the insect’s wings was traced. Analyses confirmed the presence of only one vein radius, with a clearly visible lumen.

Except for the subcostal thickening and radius vein, which are intensely pigmented, the rest of the wing membrane is semi-transparent. The mentioned subcostal thickening runs almost from the wing base and extends near the wing apex. Right under the subcostal thickening, the radial vein is present. They run parallel to each other and there is no connection between them along the entire length of the wing.

Studies showed that there is no such vein as media. The element visible under a light microscope is, in fact, only an indentation on the dorsal side of the wing membrane. The same situation applies to the anal vein. After analysis, we propose a new nomenclature for the wings within the family Matsucoccidae—only one vein radius (R), subcostal thickening (sct), median fold (med) and anal fold (af).

Another noteworthy aspect of the research was the discovery of eight campaniform sensilla on the dorsal side of the wing, in the form of a cluster. There were also two sensilla of this kind placed ventrally. In both cases, sensilla were found on the basal part of subcostal thickening.

The results of our study show the differences in venation within scale insects belonging to archaeococcoids, compared with previous studies [25]. *O. urticae* has two veins on the wing’s cross-section, while *M. pini* has only one vein. It prompts further research into how this phenomenon looks in other insects from this group. Without these studies, no general conclusions can be drawn about wing venation within the genus *Matsucoccus.*

Due to the fact that the image of the wing obtained from a light microscope gives a different impression of the venation than after the cross-section of the wing, this method allows for a more accurate interpretation of the venation of the wings of living insects. This may be the basis for determining which elements previously thought to be veins are actually veins, and which are only depressions in the wing membrane. This kind of research is necessary, especially since there is such a large discrepancy in the naming of the elements visible on the wings. The identification and proper naming of wing veins are crucial for understanding the evolutionary relationships and classification of insects. Therefore, accurate methods for interpreting wing venation are necessary, and the cross-sectional analysis method used in this study can serve as a useful tool for future research.

In summary, the study of *Matsucoccus pini’s* wing venation and sensilla provides valuable information about the diversity of wing structures in scale insects and the challenges of accurately interpreting these structures in living insects. Further research into wing structures within archaeococcoids could lead to a better understanding of the evolution of insects and their wings.

## Figures and Tables

**Figure 1 insects-14-00390-f001:**
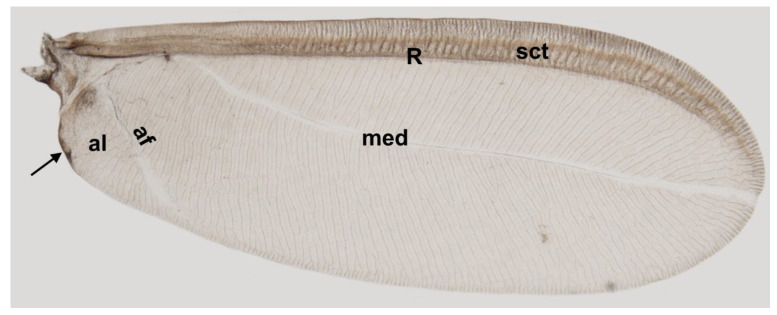
Forewing of *Matsucoccus pini* under LM. R—Radius, sct—subcostal thickening, med—median fold, al—anal lobe, af—anal fold; black arrow—narrow fold of anal lobe to hold hamuli.

**Figure 2 insects-14-00390-f002:**
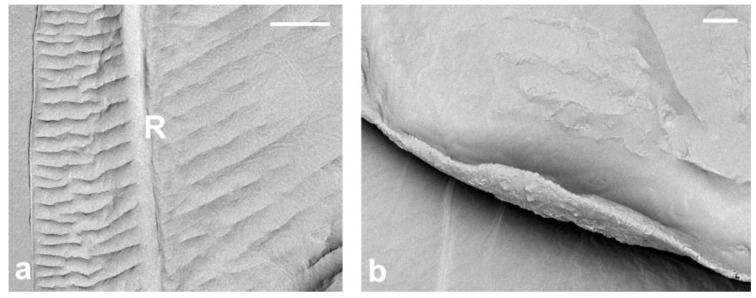
Wings of *M. pini*; (**a**) sculpture of the dorsal side of the wing membrane; (**b**) fold of the anal lobe. Scale bar (**a**) 50 μm; R—Radius. (**b**) 10 μm.

**Figure 3 insects-14-00390-f003:**
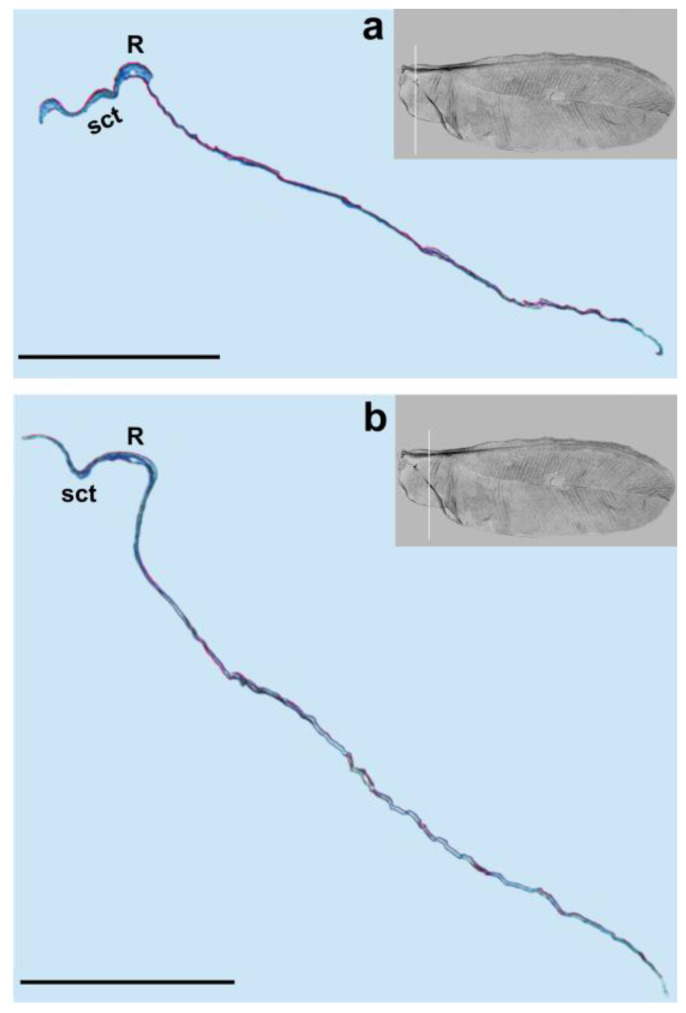
Cross-section of the wing of *M. pini*. Scale bar 100 μm, (**a**,**b**) median fold in black dotted circle.

**Figure 4 insects-14-00390-f004:**
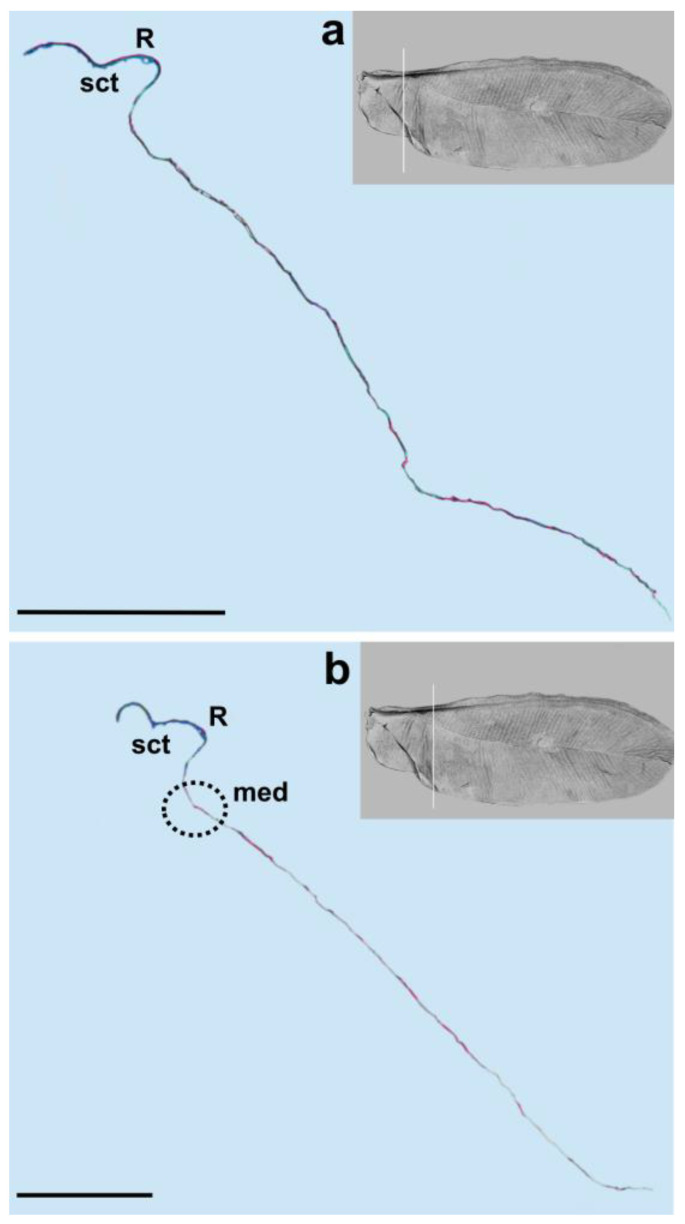
Cross-section of the wing of *M. pini*. Scale bar 100 μm; (**a**,**b**) median fold in black dotted circle.

**Figure 5 insects-14-00390-f005:**
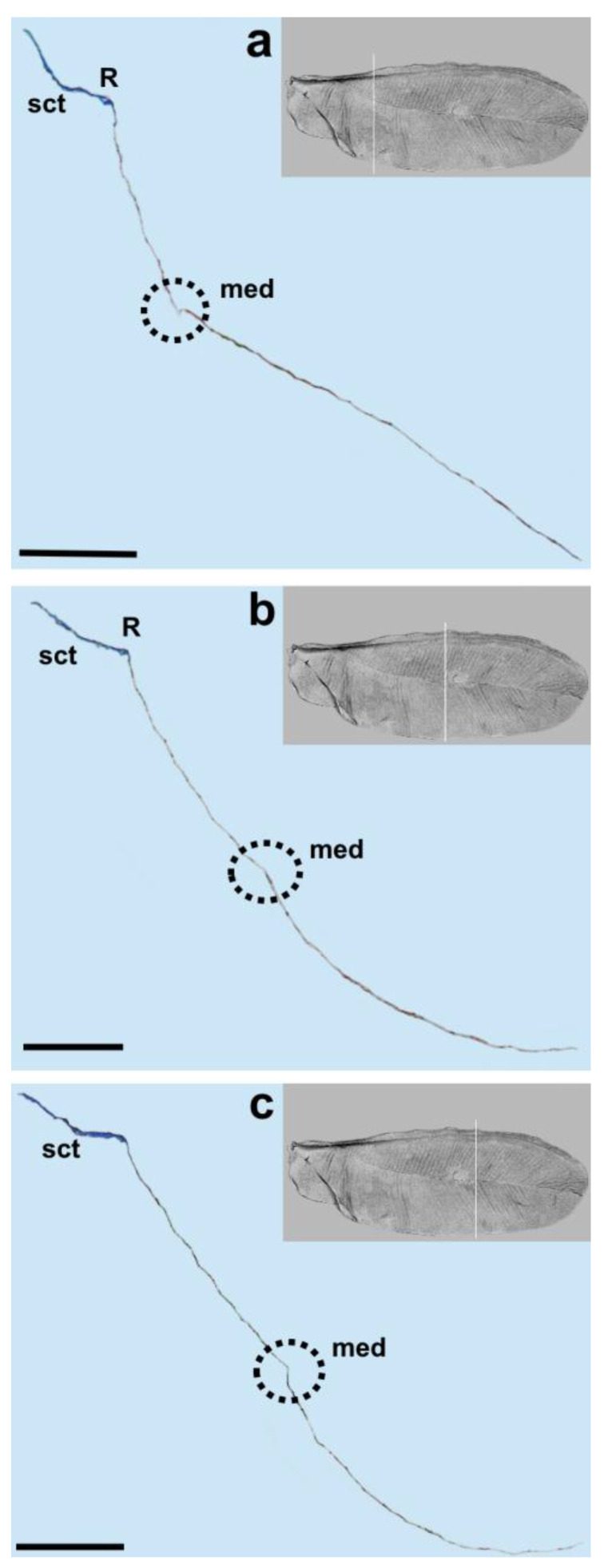
Cross-section of the wing of *M. pini*. Scale bar 100 μm; (**a**–**c**) median fold in black dotted circle.

**Figure 6 insects-14-00390-f006:**
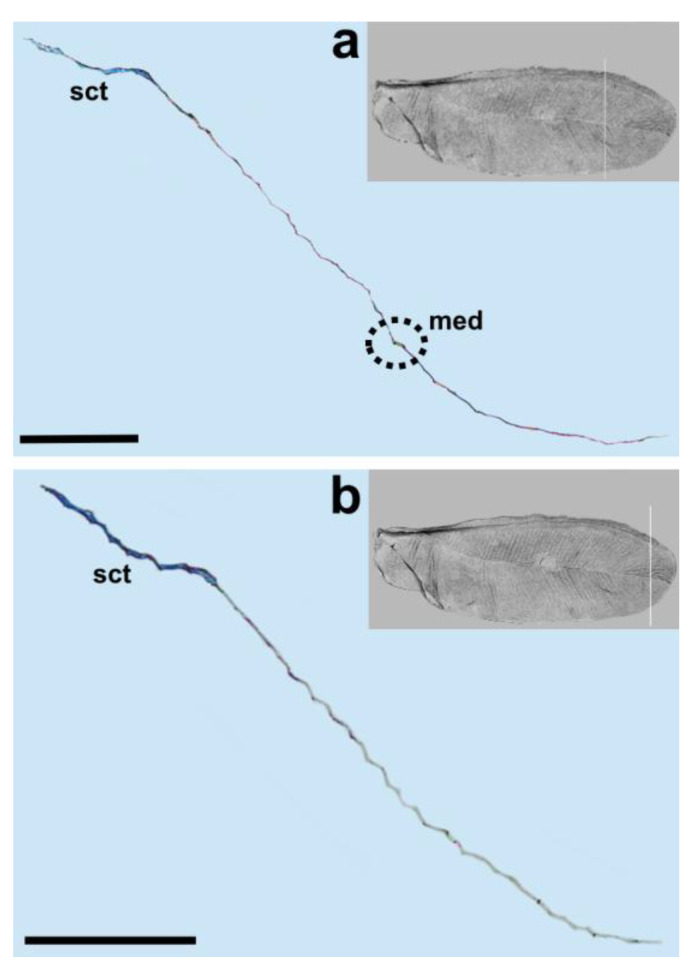
Cross-section of the wing of *M. pini*. Scale bar 100 μm; (**a**,**b**) median fold in black dotted circle.

**Figure 7 insects-14-00390-f007:**
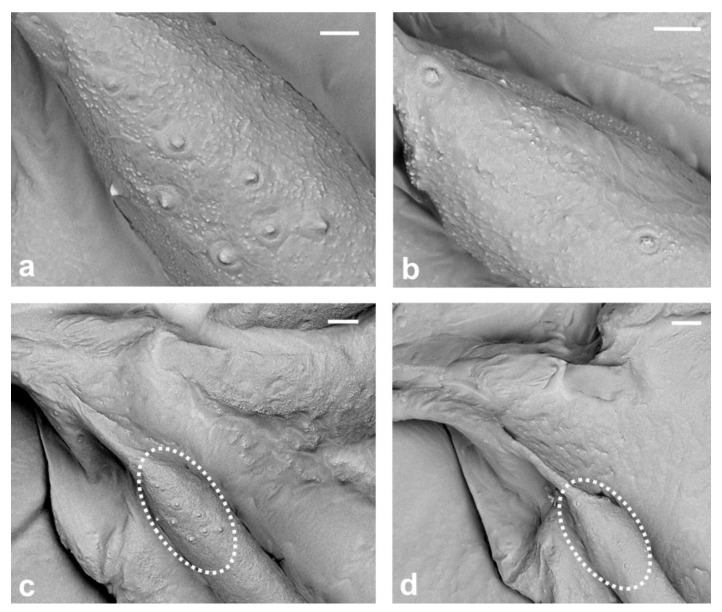
Campaniform sensilla on the wings: (**a**) dorsally; (**b**) ventrally; (**c**,**d**) location of the CS in white dotted circles. Scale bar (**a**,**b**) 5 μm and (**c**,**d**) 10 μm.

**Table 1 insects-14-00390-t001:** Wing vein nomenclature.

Author	Taxa	Veins and Lines					
Morrison (1928)	Margarodidae s.l.	costal complex (Sc + R)		Apical diagonal vein (RS)	1st diagonal light line (MS)	Basal diagonal vein (Cu + CuS)	2nd light line (1AS)
Morrison (1928)	*Matsucoccus* *matsumurae*	costal complex (Sc + R)				Single diagonal vein (Cu + CuS)	Posterior light fold
Beardsley (1968)	*Matsucoccus* *bisetosus*	subcostal thickening (sct)/s.c.	Radius (r)			Media (m)	Anal fold (af)
Siewniak (1976)	*Matsucoccus pini*	Subcosta (sc)	Radius (rad)			Media (med)	Analis (al)
Koteja (1984)	Fossil *Matsucoccus*	Subcostal thickening (sct)	Radius (r)			Media (m)	Anal vein (af)
Foldi (2005)	*Matsucoccus* *feytaudi*	Subcostal (sc)	Radial (r)			Median (m)	Anal fold (af)
Hodgson and Foldi (2006)	*Matsucoccus josephi*	Subcostal thickening (sclt)	Radius (rad)			Media (med)	Anal fold (af)
Wu and Xu (2022)	*Matsucoccus* *bisetosus*	Subcosta (Sc)	Radius (R)			Media (M)	Anal fold (Af)
Franielczyk- Pietyra et al. (2023)	*Matsucoccus pini*	subcostal thickening (sct)	Radius (R)			median fold (med)	anal fold (af)

## Data Availability

Data is contained within the article. The data presented in this study are available on request from the corresponding author. The data are not publicly available due to the large number of cross-section photos.

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
