# Peer review of "Is Every Vein a Real Vein? Cross-Section of the Wing of Matsucoccus Pini (Insecta, Hemiptera, Coccoidea: Matsucoccidae)"

_insects, 2023, doi:10.3390/insects14040390_

Round 1

Reviewer 1 Report

Is every vein a real vein? Cross-section of the wing of Matsucoccus pini (Insecta, Hemiptera, Coccoidea: Matsucoccidae).

Such Matsucoccus pini and other scientific names must be written with author name.  Also in the text the first mention of the scientific names must be mentioned with order and family name.

Scale insects (coccoids) must be Scale insects (Hemiptera: Coccomorpha)

The male of scale insects has only one pair of wing. It is better to use wing of male. instead of  forewings better to use wings only.

Otherwise the paper is well written and contains very interesting basic information.

It can be publish.

With my very best regards.

Reviewer 2 Report

The manuscript provides new interesting data on the wing morphology of Matsucoccus – the most primitive genus of the recent scale insects. The text is clear, well organized and satisfactory illustrated. The general discussion and review of the terminology of wing venation is also very useful. I advise Editorial Board to accept the manuscript for the publication.

On the other hand, there are some misunderstandings in the Introduction, concerning the phylogeny and taxonomy of scale insects. I advise authors to think about some equivocal phrases and revise these. In particular, the authors write:  The relationships within archaeococcoids remain unresolved and their monophyly is uncertain”. Meanwhile, the relationships within any group of organisms are “resolved” at the current level of knowledge only. Different researches provide their evolutional hypothesizes – phylogenetic reconstructions & taxonomic systems for reflecting the relationships, but these relationships are the subject of continuous investigations and discussions. Of course, you can agree or do not agree with the exact reconstructions & systems. For archaeococcids there were different phylogenetic reconstructions where the relationships were resolved according to the data, used by appropriate specialists. You are citing all these papers/books in your text and so, you can choice the exact reconstruction which seems more realistic according to your opinion.

It is not clear, what do you mean when write “…monophyly is uncertain”. All modern coccidologists consider archaeococcids (= superfamily Orthezioidea) as a paraphyletic group. It means that archaeococids are the ancestor group for neococcids. Evolutional taxonomic school accepts and widely uses paraphyletic taxa, whereas cladistic taxonomic school rejects paraphyletic taxa.

Please, see also some other comments directly in the PDF file of the manuscript.

Reviewer 3 Report

Dear authors, dzień dobry,

Read with interest your Ms.  Here my report:

The manuscript “Is every vein a real vein? Cross-section of the wing of Matsucoccus pini (Insecta, Hemiptera, Coccoidea: Matsucoccidae)” by Dr Barbara Franielczyk-Pietyra and colleagues presents images of the wing of males of a scale insect and discuss the nomenclature used by others.  They show that this genus has only one alar vein.  The ref list looks comprehensive.  This Ms is interesting, though not for a wide audience, mainly for coccoid taxonomists and anatomists (what I am not). 

As a non-native English speaker, I cannot judge of the quality of the language, even though I reported a few mistakes or made some suggestions below.  The Ms should be revised by a native English speaker. 

Many paragraphs, sometimes consisting of only one sentence or line, could be fused.

Finally, this Ms could, in my opinion, be accepted with minor modifications.  More specific comments follow.

Specific comments

The title should mention “males”.

Line 17: “series” or “cluster” rather than “row”

L 25: the superfamily

L 27: Sternorrhyncha taxa

L 34: mostly

L 42: into head

L 54: Northern Hemisphere

L 75: Only with cross-sections is there a…

L 89: insert commas after XL and BV

L 93: Petri

L 98: OsO4 – 4 as subscript

L 100-101 cut from root … with a diamond knife on a Leica..:

L 104: Fig. x?

L 113: “al” versus “af” are somewhat ambiguous

Fig. 2a should be better described in the caption: posterior/anterior? dorsal/ventral view?  Add “R” on the Fig.

L 126: backwards rather that downwards

Figs 3-6: the lettering, the circle and the vertical bar are poorly visible.  Why not in black?

L 145: for a short distance from the base

L 147: distal rather than last; band rather than patch

L 152: remove “the” and the 2nd “are absent”

L 151: move upwards the terms between (…), since is doesn’t refer to hamuli

L 164: We present also…

Table: Morrisson’s structures RS and MS are not discussed in the text.

L 178+: in the present study

L 186: suggested (= not proven) or observed?

L 203-206: explain a bit.  Is the presence of 2 veins specific of genus Matsucoccus or family?

L 212: mention family of Xylococcus (see also L 245)

L 222: (Fig. 1)

L 243: only one cluster (I presume)

L 245: considered closely related to…

L 248: mention family of Stigmacoccus spp.

L 253: taxa rather than groups (also L. 273)

L 255: considered rather than concerned

L 257: microtrichia was not observed either

L 262: a pterostigma

L 262-265: what about neococcoids?

L 268: extant rather than recent

L 271: remove considerably?

L 338: lub?
